# Cardiovascular Safety and Benefits of Testosterone Implant Therapy in Postmenopausal Women: Where Are We?

**DOI:** 10.3390/ph16040619

**Published:** 2023-04-20

**Authors:** Guilherme Renke, Francisco Tostes

**Affiliations:** 1Nutrindo Ideais Performance and Nutrition Research Center, Rio de Janeiro 22411-040, Brazil; 2Clementino Fraga Filho University Hospital, Federal University of Rio de Janeiro, Rio de Janeiro 21941-617, Brazil

**Keywords:** menopause, implant, testosterone, hormone replacement therapy

## Abstract

We discuss the CV safety and efficacy data for subcutaneous testosterone therapy (STT) in postmenopausal women. We also highlight new directions and applications of correct dosages performed in a specialized center. To recommend STT, we propose innovative criteria (IDEALSTT) according to total testosterone (T) level, carotid artery intima-media thickness, and calculated SCORE for a 10-year risk of fatal cardiovascular disease (CVD). Despite all the controversies, hormone replacement therapy (HRT) with T has gained prominence in treating pre and postmenopausal women in the last decades. HRT with silastic and bioabsorbable testosterone hormone implants has gained prominence recently due to its practicality and effectiveness in treating menopausal symptoms and hypoactive sexual desire disorder. A recent publication on the complications of STT, looking at a large cohort of patients over seven years, demonstrated its long-term safety. However, the cardiovascular (CV) risk and safety of STT in women are still controversial.

## 1. Introduction

The production of androgenic hormones in women decreases markedly in the first reproductive years [1]. At 40 years of age, a woman has a 50% reduction in plasma total testosterone (T) levels compared to a 21-year-old woman [2]. Symptoms of androgen deficiency may rarely affect women in the reproductive phase. However, in postmenopausal women, these symptoms can be devastating. This includes decreased memory, sense of well-being and libido, reduced muscle mass and bone mass, increased irritability, anxiety, fatigue, and symptoms of depression [1,3,4]. Subcutaneous testosterone therapy (STT) administered by silastic or bioabsorbable implants has been used successfully in women since the 1930s. Published studies demonstrate promising results and safety in doses ranging from 75 to 225 mg [5,6,7,8,9]. Thus, higher doses of T (500–1800 mg) in subcutaneous implants have been used effectively to treat patients with breast cancer [10,11]. Because it is not excreted in breast milk, T has been used to treat symptoms of postpartum depression and fatigue during lactation and the puerperium [12]. T therapy alone, at physiological doses, has been reported to be more effective than estradiol (E2)-T therapy or E2 alone for relieving postmenopausal symptoms with the same cardiovascular (CV) safety [13].

In this perspective, we summarize whether it is prudent to develop an approach for managing postmenopausal women through STT, evaluating its possible CV benefits and risks.

### 1.1. The Controversy of T Therapy in Women

For over 60 years, hormone replacement therapy (HRT) with T has been used to treat perimenopausal and menopausal symptoms [14]. In some countries such as Brazil, T therapy has been approved for use in women with hypoactive sexual desire disorder [15]. The rational use of androgens in women’s health and well-being is of great interest. Around the world, thousands of women report that T therapy has brought about improvements in their quality of life and overall health. Despite these data, there is still much controversy about the existence of androgen deficiency, its diagnosis, and clinical treatment. Unfortunately, androgen use is uniquely associated with masculinity or male sexual function. This has undoubtedly contributed to the lack of recognition of the effects of low-dose androgen therapy use in women. Androgens are essential for women in both female reproductive function and hormonal balance and as vital precursors in synthesizing female hormones such as E2 [14].

Female androgen insufficiency was defined and classified with the release of the Princeton Consensus in 2002 [16]. Thus, problems related to T hormone deficit in women were evaluated for the first time. However, the major criticism is the lack of a correct determination of normal laboratory T levels in women. Current studies on the serum levels of this steroid could be more satisfactory, mainly due to the lack of sensitivity in determining the normal laboratory range. Additionally, the consensus statement recommended that balance dialysis methods be used for the most accurate assessment of bioavailable T in women. As increased levels of sex-hormone-binding globulin (SHBG) can also cause androgen insufficiency even in the presence of normal levels of total T, it was recommended to calculate and evaluate the free T index for this purpose in addition to the clinical symptoms of the patient [16].

In 2006, the Endocrine Society published a guideline against androgen therapy in women [17]. This position was based on the absence of well-defined criteria for diagnosing androgen deficiency in women. Another critical point was the lack of safety studies of long-term androgen administration in women and the certainty of the correlation between sexual symptoms and plasma T levels. In addition, as there were few clinical studies at the time, the guideline recommended that more studies be conducted to indicate the use of T in treating androgen deficiency.

However, many endocrinologists disagreed with the statement of the Endocrine Society Clinical Practice Guidelines on androgen therapy [17]. Notably, the guideline ignored studies that compared T levels in healthy premenopausal women with hypoactive sexual desire disorder (HSDD) and defined normal ranges for calculated free T [2,18,19]. Furthermore, although no correlation was found between total T levels and sexual dysfunction, there was a correlation between low levels of dehydroepiandrosterone sulfate (S-DHEA) and HSDD [20]. Thus, the unreliability of the results was due solely to the difficulty in accurately measuring free T levels in women. The argument that there was a lack of data on the safety of long-term T administration in women was refuted [17], stating that several studies showed only minor skin changes (oily skin and acne) and that the beneficial effects of the use of T (sexual enhancement, increased libido, and quality of life) should receive more significant evidence [21,22,23,24]. Therefore, because of these events, we observe that the use of T for treating HSDD in women remains controversial.

### 1.2. The Challenge of Low T Diagnosis in Women

Laboratory analysis of T in women is controversial. An average blood level of T and what laboratory parameters should be used to guide clinicians in treatment still need to be improved. The diagnosis of T deficiency is usually made by analyzing the serum levels of total T, which is not the best parameter for evaluation in women. There is no standard analysis for women since much of the total circulating T is biologically unavailable due to its binding to sex hormone-binding globulin (SHBG). Bioavailable T better represents a woman’s androgenic status [1]. Therefore, the calculation of bioavailable T is one of the best parameters for evaluating the androgenic profile in women as it disregards T values linked to SHBG [14].

A complete hormonal assessment should be performed when T deficiency is suspected, including total T, SHBG, free T, and E2. Due to the low T levels in women compared to men, precision limitations exist on the commonly available assays used to measure T. The most widely used assay to measure total T in women is the radioimmunoassay (RIA) [25]. This assay has not been standardized to measure T levels in women and may result in unreliable T levels. However, LC-tandem mass spectrometry or liquid chromatography/gas chromatography assays can obtain reliable and accurate results to measure total T. The reference ranges of total T in women are 45.5–57.5 ng/dL, 27.6–39.8 ng/dL, and 27.0–38.6 ng/dL for 20–29 years, 20–39 years, and 40–49 years, respectively [1,14,25].

Although there is no diagnostic value for T deficiency in women, it is recommended not to prescribe T for postmenopausal women with signs of increased androgens. This includes the presence of clinical symptoms (acne, androgenic alopecia, hirsutism, etc.) or normal T levels (medium or high ranges) [23].

The standard of care is always to evaluate and treat the clinical syndrome of T deficiency that is not linked to specific laboratory values. Most specialists use therapy based on the calculation of bioavailable T. However, the goal should be to individualize dosing to resolve clinical symptoms while keeping serum levels low enough to minimize possible T side effects. Successful results have been observed with STT in reducing symptoms with minimal side effects by maintaining serum T levels between 150 and 250 ng/dL in women. However, additional long-term dosing studies should be conducted to evaluate T deficiency in women [1,14].

### 1.3. Monitoring T Therapy in Women

Each type of T hormone implant, bioabsorbable or silastic, has different rates of release and absorption, which may alter the pharmacokinetics of its effect on androgen receptors. Due to this variability, depending on the type of implant used, constant monitoring is the key to safety and effective treatment. The purpose of monitoring is to ensure proper dose titration and prevent overdosing.

Before initiating HRT, an assessment of lipid profile and fasting liver function tests must be obtained and monitored annually. In addition, hyperlipidemia and hepatic dysfunction are contraindications for T therapy. After starting STT, a complete blood sample should be reassessed at two months and repeated within 6–8 weeks whenever there is a dose adjustment [14,25]. As the effect of therapy is progressive, patients should be advised that seeing maximum results will take up to 12 weeks. Despite this, some women see symptomatic improvement 4–6 weeks after the start of treatment [14].

### 1.4. The Evolution of STT in Clinical Practice

Some ways to prescribe HRT with T include transdermal gel, oral or sublingual tablets, intramuscular injections, and the application of silastic implants or bioabsorbable pellets [1]. Implants were first described as an alternative treatment for menopausal symptoms in the 1950s [26]. Studies demonstrate that this route of administration provides a safety profile and effective therapy without the fluctuations in blood levels commonly seen with transdermal administration or intramuscular injection [27]. Thus, STT administration has been used worldwide for decades. In some countries, such as the United States of America (USA) and Brazil, non-absorbable (silastic) and absorbable implants (pellets) are widely used. Another peculiarity of implants is the absence of peaks in serum T levels, reducing the possibility of side effects and secondary reactions [28]. An extensive 7-year retrospective study of 400.000 women and over 1.200.000 subcutaneous implant procedures performed shows a safety profile. The overall complication rate was <1%, and the continuation of treatment after two insertions were 93% (C.I. 90–95) [14].

Therapy with bioabsorbable T implants is composed, in an outsourced pharmacy, of bioidentical T (USP) and steric acid (5.21%). A Food and Drug Administration (FDA)-approved process compresses the substrates into 3.1 mm (diameter) cylinders, sealed in glass ampoules, and sterilized by E-beam sterilization. [29] Sterile implants with dosages from 50 to 75 mg are inserted, through a 5 mm incision, into the subcutaneous tissue of the upper gluteal area or lower abdomen through a small anesthetized incision using a sterile stainless-steel trocar or disposable trocar kit in a simple 5 min procedure. This practice, more common in the USA, increased the possibility of using long-acting HRT every 12 to 16 weeks. The advantage of bioabsorbable implants is that they dissolve completely and do not need to be removed. However, they have the disadvantage of the impossibility of withdrawing them in case of need. [1]

Silastic implants are also composed, made in an outsourced pharmacy (ELMECO, Salvador, Brazil), of bioidentical T (USP) in tubes of dimethylpolysiloxane (silicone). To manufacture implants, segments 4 cm in length and 2.4 mm in diameter are used. Pieces of silastic tubes are filled with 40 mg of T and sealed at both ends with medical silicone adhesive [30,31,32,33]. T is estimated to be released from the implant capsule at approximately 110 mcg daily and has an average duration of 38–48 weeks. After the end of treatment, implants may remain in the patient. However, removing silastic implants is recommended, especially if the treatment is continued, which should be performed by a specialized medical team. In clinical practice, we have observed that subcutaneous T is consistently absorbed and clinically more effective than transdermal T use.

### 1.5. T Impact on Cardiovascular Health

Although T is the primary sex hormone in men, it has multiple critical physiological functions. For example, it is essential in the central nervous, musculoskeletal, and CV systems. Moreover, T is necessary for carbohydrate, lipid, and protein metabolism [34,35]. T participates in metabolic homeostasis, and its deficiency is usually found in men with metabolic syndrome. Previous studies have consistently demonstrated that statement. Hypogonadism predisposes men to endothelium dysfunction, inflammation, insulin resistance, obesity, abnormal lipid profiles, and borderline or overt hypertension [36]. A meta-analysis by Araujo et al. included 12 studies involving more than 17.000 participants and revealed that low endogenous T levels were associated with overall and CV mortality [37].

The cardioprotective effect from E2 and possible deleterious action from T on the CV system is supposed to justify the higher rates of coronary artery disease (CAD) and mortality [38]. However, the physiological concentration of T tends to be cardioprotective in males [39]. Indeed, T, a potent vasodilator through calcium antagonistic action, has been reported to be protective against angina, especially in men with a low T level [40]. Consistently, previous studies have shown an association between a low T level and an increased risk of CV diseases [38,41,42] and a lower T level in men with CAD compared to those without [43,44]. Men with a low T level are more likely to have atherosclerotic plaques, endothelial dysfunction, and higher levels of high-sensitivity C-reactive protein (CRP) [45].

In what concerns women, there has been no association between endogenous levels of T in women and CAD [46], and other studies [47,48,49,50] showed that it might have a protective effect, as demonstrated by a meta-analysis of randomized controlled trials [51]. Studies that observed women for up to 2 years showed that, with T levels at the upper portion or slightly above the reference range for reproductive-aged women, there is no increase in adverse CV effects, including changes in blood pressure, blood viscosity, arterial vascular reactivity, hypercoagulable states, and polycythemia [52,53,54,55].

Although models of supraphysiologic T levels, such as polycystic ovarian disease (PCOS), have shown an increased risk of presenting CAD. The mechanism of hyperandrogenemia in PCOS is believed to be secondary to hyperinsulinemia. Increased CV risk is related to metabolic syndrome compounded by visceral obesity, insulin resistance, dyslipidemia, and hyperandrogenemia [56]. Exogenous oral and transdermal T studies at physiologic levels do not lead to insulin resistance or changes in fasting glucose [57,58,59,60].

Two prospective studies demonstrated a lower risk of cardiovascular disease (CVD) in women with endogenous T levels at the higher reference quintiles [61,62]. Studies of different types of T administration (subcutaneous implant, transdermal patch, spray, or gel) do not present worsening in lipid levels, CRP, glycated hemoglobin, or insulin sensitivity. In a large randomized controlled trial, the transdermal T patch did not increase any adverse effects on lipids, insulin resistance, and CRP in postmenopausal women compared to the placebo [63]. A smaller randomized controlled trial (RCT) of premenopausal women showed no adverse effects of a transdermal T spray versus placebo [64]. Moreover, T replacement improved vasodilatation parameters, such as brachial artery flow-mediated dilatation (FMD) in postmenopausal women already receiving E2 [65].

An RCT compared women with documented congestive cardiac failure who received T therapy through a patch releasing 300 mg of T/day with women treated with a placebo. The T treatment group women improved in several functional tests, such as peak oxygen consumption (VO2 max), distance walked over the 6 min walking test, muscle strength, and insulin resistance [66]. The available database shows no evidence to support that T treatment in women, at physiological levels or slightly above, increases any adverse effects related to CVD risk [67,68].

The assessment of CV risk in the latest European Society of Cardiology (ESC) and European Atherosclerosis Society (EAS) [69] shows the probability of a person developing an atherosclerotic CV event over a defined period. The absolute CVD risk expresses the combined effect of several risk factors in this risk estimate. SCORE data indicate the absolute risk of cardiovascular events that are divided into low risk (calculated SCORE <1% for 10-year risk of fatal CVD), moderate risk (calculated SCORE > 1% and <5% for 10-year risk of fatal CVD), or high risk (calculated SCORE > 5% and <10% for 10-year risk of fatal CVD). Furthermore, examining the measurement of the carotid artery intima-media thickness percentiles and carotid plaque burden with ultrasound has been demonstrated to predict CV events [69].

### 1.6. The Benefits and Effectiveness of T Implant Therapy

SST has been used in women since the 1930s in doses ranging from 40 to 240 mg. Indeed, long-term data exist on these dosages’ safety, efficacy, and tolerability over more than 35 years of therapy [7,10,70,71,72,73,74]. Furthermore, studies with supraphysiological T doses that treat special conditions, such as breast cancer and transgender patients, have already been studied and found safe [13,45,74,75,76,77,78]. We summarized the benefits, possible applications, and STT doses used in past and recent trials (Table 1).

Dosage calculation for the long-acting, sustained-release T silastic implant is based on the patient’s weight and body composition. This is consistent with other studies reporting that the T effect is dose-dependent, so choosing the correct dose can make all the difference in treatment [13,24,45,61,75,78,79]. Despite this, in our clinical experience of more than 5000 procedures, we consider it safe to use low dosages (40 to 120 mg) with the same effectiveness as the high doses. However, as previously published, the T doses usually used (40–240 mg) are clinically effective and well tolerated [1,24,45,61,80]. Higher doses of T have been shown to correlate with a more significant improvement in quality of life. In addition, previous studies have not confirmed side effects such as deepening of the voice [81] and worsening of cholesterol [82]. There is an improvement in symptoms with T implants as evidenced by the ‘Menopause Rating Scale’ total score, including symptom reduction and improvement in psychological and urogenital complaints [1].

**Table 1 pharmaceuticals-16-00619-t001:** Safety and benefits of STT in women. MRS: Menopause Rating Scale, T: testosterone, E2: estradiol.

Author(s), Year, Reference Number	Implant Dose	Characteristic of Study/Participants	Results
Glaser et al. (2011) [1]	Subcutaneous T implanted ranged between 75 mg and 160 mg.	Prospective study (300 pre and postmenopausal women with symptoms of relative androgen deficiency)	Improvement in total MRS score, as well as psychological, somatic, and urogenital subscale scores.
Burger et al. (1984) [6]	Combined subcutaneous implants with E2 (40 mg) and T (100 mg)	Prospective study (17 postmenopausal women)	Substantial symptomatic relief, particularly in libido, while causing rises in mid-follicular concentrations of E2 and maximal T levels about three times normal, without significant effects on plasma lipids.
Gambrell et al. (2006) [7]	Combined subcutaneous implants with E2 (25 to 100 mg) and T (75 to 225 mg)	Prospective study (606 postmenopausal women)	Menopause-symptom-relieving, side-effect-free regimen, treatment continuation rates of 72.1% for 5 years and 53.5% for 10 years.
Garnett et al. (1991) [8]	Combined subcutaneous implants with E2 (50 to 75 mg) and T (100 mg)	Cross-sectional study (110 postmenopausal women)	Subcutaneous E2 and T prevent postmenopausal osteoporosis and maintain normal bone density for as long as treatment is continued.
Thom et al. (1981) [9]	Combined subcutaneous implants with E2 (50 to 100 mg) and T (100–200 mg)	Prospective study (24 postmenopausal hysterectomized patients)	Plasma T concentrations rose from a mean concentration of 28.89 ng/dL to 144.45 ng/dL and 193 ng/dL after implants of 100 mg and 200 mg of T, respectively.
Dimitrakakis et al. (2004) [10]	Combined subcutaneous implants containing T (50 mg to 150 mg) with conventional E2 or E2 plus progestin treatment.	Prospective study (508 postmenopausal women)	The addition of T to conventional hormone therapy for postmenopausal women does not increase and may indeed reduce the hormone-therapy-associated breast cancer risk.
Donovitz et al. (2021) [29]	T implants (127.2 ± 17.7 mg) or T (127.6 ± 28.7 mg) plus E2 (15.6 ± 6.5 mg)	Retrospective study (2377 postmenopausal women)	Subcutaneous T alone or associated with E2 implants significantly reduced the incidence of breast cancer in pre and postmenopausal women
Glaser et al. (2019) [70]	T implant dosing was weight-based with an average dose of 2–2.5 mg/kg (minimum dose 120 mg)	Prospective study (1267 pre and postmenopausal women)	10-year therapy with subcutaneous T, or T combined with anastrozole, did not increase the incidence of breast cancer
Glaser et al. (2012) [76]	T implants 130 ± 19.7 mg (range 100–160 mg)	Prospective study (27 pre and postmenopausal women previously diagnosed with migraine headache)	Continuous T implants was effective therapy in reducing the severity of migraine headaches in both pre and postmenopausal women
Glaser et al. (2016) [81]	T implant dosing was weight-based with an average dose of 2–2.5 mg/kg (mean T dose was 138 ± 22.7 mg)	Prospective study (10 postmenopausal women)	Subcutaneous T had no adverse effect on the female voice including lowering or deepening of the voice.

Additionally, these T implant doses have increased scalp hair growth and are not associated with androgenic alopecia [78]. As expected, most patients had a concomitant increase in facial hair growth. However, no patients discontinued therapy due to increased hair growth. A similar dosage ranging from 80–200 mg can treat migraine headaches in pre and postmenopausal women and is safely used in breast cancer survivors to treat symptoms of T deficiency [1,76]. There were no adverse events related to subcutaneous T therapy; besides increased facial hair and mild acne, these doses had minimal side effects. It has been documented that serum T levels on STT are more significant than endogenous ranges [1,14,70,71,76]. More consistent benefits are seen with T levels exceeding the normal range [77].

Despite the benefits and excellent clinical experience, we must admit that some barriers exist to STT, such as prejudice from patients or doctors, high treatment costs, and lack of training for removing implants. In addition, despite the benefits of post-menopause, long-term safety studies using T implants in premenopausal women to treat HSDD still need to be improved.

### 1.7. T Implant Therapy and Breast Cancer Risk

The most common cancer in women is still breast cancer, so preventive strategies are essential from a public health point of view. Evidence supports the protective effect of T therapy on breast tissue [29,70]. The long-term impact of biosimilar T therapy showed no increase in breast cancer incidence. T therapy is related to a lower risk of breast cancer in pre and postmenopausal women [70].

In recent years, two extensive studies, Glacer et al. (2019) [70] and Donovitz et al. (2021) [29], looked at the impact of STT and cancer risk in a total of 3644 pre and postmenopausal women. In the first study, a 10-year prospective cohort investigating the incidence of breast cancer in women treated with STT showed a 39% reduction in the incidence of breast cancer in the studied population compared to the expected incidence of Surveillance Epidemiology and End Results (SEER) for the same age. STT used to treat symptoms of T deficiency in pre and postmenopausal women did not increase the incidence of invasive breast cancer [70].

The second retrospective study, with a 9-year follow-up, demonstrated a reduction of 35.5% in breast cancer incidence in both T and T and E2 hormone implant users compared to age-specific SEER incidence rates (223/100,000) [29]. A reduction in breast cancer cases was seen in the T implant group compared to the expected number of breast cancer cases corresponding to SEER age (14 vs. 48 cases). The breast cancer incidence rate was 144/100,000 for the T therapy group vs. 223/100,000 for the SEER data. If the results were compared with data from the Women’s Health Initiative (WHI) placebo arm, there would be the same result: a marked reduction in breast cancer cases with the T implant (14 vs. 71) compared with the placebo arm of the WHI [83].

### 1.8. New Directions and Future Approaches in STT

Significant barriers still exist to the use of T implants. The first refers to the use of T itself. The major criticism is the absence of diagnostic criteria and therapeutic doses established in pre and postmenopausal women. The second point, already related to the implants, refers to the lack of standardization of therapeutic doses and safety studies on the hormone release rate from implant devices. A significant concern also exists regarding the excessive or inappropriate use of HRT, especially in high dosages. However, despite the conservative position of the different bodies (Endocrine Society in the US and SBEM (Brazilian Society of Endocrinology and Metabolism) in Brazil) regarding hormonal implants, many physicians criticize or disagree with this stance. Many specialists worldwide (endocrinologists and gynecologists) defend the correct and consistent use of STT therapy with the dosages already discussed and used in numerous long-term clinical studies that have reached more than 4000 women pre- and post-menopause, demonstrating the satisfactory safety and effectiveness of this treatment.

Our clinical experience at Nutrindo Ideais Performance and Nutrition Research Center has reached almost ten years of the use of therapies with STT. Our records show it has been used for more than 2500 treated patients with preliminary data on continuation rate, complications, and therapeutic benefits similar to those published in recent studies [84]. Thus, we propose and recommend some criteria (IDEALSTT) that we consider innovative according to our great clinical expertise and previous trials (Table 2).

To consider the therapy, we recommend the following: (1) Use a minimum dosage of 100 mg in the initial STT. (2) Use the maximum dosage of 2.5 to 3 mg/kg in STT. (3) Despite the more significant number of studies with absorbable implants, we recommend using silastic implants due to the absence of peak or significant initial hormonal release, the absence or lower risk of extrusion, and the possibility of interruption or removal of the implants. (4) Do not start the therapeutic approach of HRT with the STT; always perform a therapeutic test with transdermal gel before using the implants. (5) Consider maintaining blood total T between 50 and 250 mg/dL.

Regarding CV risk, T therapy data in women are still scarce. However, HRT with T reduces numerous CV risk factors, including insulin resistance, visceral fat reduction, and body composition improvement [60,61,62,63,64,65,66,67]. Thus, further randomized studies with STT are necessary to evaluate the CV risk in women.

## 2. Conclusions

Our findings show that the CV safety of STT with a minimum dose of 100 mg and a maximum dose of 3 mg/kg is adequate when treating women, presenting few side effects if performed correctly. To consider STT therapy, we propose and recommend innovative criteria (IDEALSTT) according to the total T level, carotid artery intima-media thickness, and calculated SCORE for a 10-year risk of fatal CVD. There may be a secondary CV benefit with T therapy in women. STT dosages should be individualized, respecting the woman’s weight and symptomatology more than laboratory levels. More extensive randomized clinical trials are needed to evaluate the long-term safety of this treatment.

## Figures and Tables

**Table 2 pharmaceuticals-16-00619-t002:** Subcutaneous testosterone therapy CV risk assessment protocol (IDEALSTT). * Must include low T symptoms: decreased memory, sense of well-being and libido, reduced muscle mass and bone mass, increased irritability, anxiety, fatigue, and symptoms of depression. CVD: cardiovascular disease.

Recommend *	Consider with Caution *	Avoid
Total testosterone level	Total testosterone level	Total testosterone level
<45.5 ng/dL (20–29 years) <27.6 ng/dL (20–39 years) <27.0 ng/dL (40–49 years)	>45.5 ng/dL (20–29 years)	>57.5 ng/dL (20–29 years)
	>27.6 ng/dL (20–39 years)	>39.8 ng/dL (20–39 years)
	>27.0 ng/dL (40–49 years)	>38.6 ng/dL (40–49 years)
--------------and----------------	---------------or-----------------	---------------or-----------------
Carotid artery intima-media thickness < 75th percentiles and the absence of carotid plaque	Carotid artery intima-media thickness > 75th percentiles and the absence of carotid plaque	Carotid artery intima-media thickness > 75th percentiles or the presence of carotid plaque
-------------and----------------	---------------or-----------------	---------------or-----------------
Calculated SCORE < 1% for 10-year risk of fatal CVD	Calculated SCORE ≥ 1% and <5% for 10-year risk of fatal CVD	Calculated SCORE ≥ 5% and <10% for 10-year risk of fatal CVD

## Data Availability

The datasets used and/or analyzed during the current study are available from the corresponding author upon reasonable request.

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
