# Peer review of "Cardiovascular Safety and Benefits of Testosterone Implant Therapy in Postmenopausal Women: Where Are We?"

_pharmaceuticals, 2023, doi:10.3390/ph16040619_

Round 1

Reviewer 1 Report

The present work evaluated the сardiovascular safety and benefits of testosterone implant therapy in postmenopausal women. In my opinion, this is a very interesting, exiting, and very useful article for this field of the science. I have some recommendations how to improve this article. A moderate correction of English language is needed. There are a some of technical and grammatical mistakes in the whole text.

 1.    Please, give a total table of all studies that are mentioned in your article concerning the type of trial and dose of testosterone application. Also, please divide all these women by age.

2.    Please, give a subsection concerning new directions of such investigations in women in the future.

3.    Please, give a total scheme of possible application of such therapy.

4.    The section “Conclusion” is not fully understand. The clarification for all points of view is needed.

In conclusion, the article is not possible to publish at least at the present form.

Author Response

C/o Editorial Office

Pharmaceuticals MDPI

April 12, 2023

Dear Editorial Office:

The submitted manuscript entitled Cardiovascular safety and benefits of testosterone implant therapy in postmenopausal women: where are we? by Renke et al. has been re-evaluated and has undergone revisions requested and suggested by the reviewers. The manuscript and new tables were sent with the corrections and adjustments flagged as "revision" in the Word reviewer.

The authors would like to respond to all reviewers' comments as follows promptly:

REVIEWERS 1 COMMENTS

“The present work evaluated the сardiovascular safety and benefits of testosterone implant therapy in postmenopausal women. In my opinion, this is a very interesting, exiting, and very useful article for this field of the science. I have some recommendations how to improve this article. A moderate correction of English language is needed. There are a some of technical and grammatical mistakes in the whole text.”

A0: We thank the reviewer for drawing attention to this point and for the opportunity to clarify it. We agree with the reviewer as follows.

  • Please, give a total table of all studies that are mentioned in your article concerning the type of trial and dose of testosterone application. Also, please divide all these women by age.”

A1: We thank the reviewer for drawing attention to this point and for the opportunity to clarify it. We agree with the reviewer; "table 1” with all studies we considered relevant to this perspective was included.

  • Please, give a subsection concerning new directions of such investigations in women in the future.

A2: We thank the reviewer for drawing attention to this point and for the opportunity to clarify it. We agree with the reviewer, and the subsection “1.8. New Directions and future approaches in STT” was included as requested.

  • Please, give a total scheme of possible application of such therapy.

A3: We thank the reviewer for drawing attention to this point and for the opportunity to clarify it. We agree with the reviewer, and subsection “1.6. The benefits and Effectiveness of T implant therapy” was improved, and all possible studied applications are also described in “Table 1”.

  • The section “Conclusion” is not fully understand. The clarification for all points of view is needed.

A4: We thank the reviewer for drawing attention to this critical point and for the opportunity to fix it. We agree with the conclusion was improved with major findings.

We hope you find our manuscript suitable for publication and look forward to hearing from you in due course. We are gratefully and appreciate the reviewers' suggestions and remain available for further adjustments and corrections.

Sincerely,

Guilherme Renke MD

Tel: +5521994431653

Email: renke@renke.com.br

Reviewer 2 Report

Pharmaceuticals

COMMENTS TO THE EDITOR AND THE AUTHORS

Manuscript ID 2289239: “Cardiovascular safety and benefits of testosterone implant therapy in postmenopausal women: where are we?"

Dear Editor and Authors,

Please find below some of the comments.

A SUMMARY OF THE CONTENT

The authors stated that they discuss the cardiovascular safety and efficacy data for subcutaneous testosterone therapy in postmenopausal women within the correct dosages and performed in a specialized center.

THE OVERALL OPINION OF THE MANUSCRIPT

The strengths: the manuscript is within the scope of the journal; the review provides some new insights about the cardiovascular safety and benefits of testosterone implant therapy in postmenopausal women.

The limitations: The text is not focused on the aim of the manuscript; the abstract is not adequately written; the original, and important pioneered results, as well as recent advance in the field focusing on the subject of the study are not cited and they are not discussed; the text is not precisely written.

COMMENTS

(1) ABSTRACT

Please rewrite the text of the abstract to be more focus on the subject/the title. At the present form, only last sentence describes the idea of the manuscript. All other sentences describe very well-known findings. Please consider to start the abstract with the aim of the study.

(2) GENERAL

(2-1) The significant amount of the text describes the very well-known and general knowledge. Please focus on the subject. Please consider rewriting the text to focus on “perspectives”; not to very well-known knowledge. It will increase interest of readers and “visibility” of the manuscript.

(2-2) Please cite the original (instead of review), and important pioneered results, as well as recent advance in the field focusing on the subject of the study. The manuscript is aimed to be “perspective”. At the present form it is not.

(2-3) Please list rules of all related agencies (not only FDA and Brazilian).

(2-4) Please be precise in the description. One of the examples is “estrogen” (line 179: “The cardioprotective effect from estrogen and possible deleterious action from T on the…”). It is very well known that there are three main estrogens (estradiol, estriol, estrone). Accordingly, please replace “estrogen” with “estrogens”. Please go through the text and please be precise in the description.

(2-5) Please remove Figure 1 since it presents very well knowledge. Please remove the text related to Figure 1. Please keep in mind that you chose to write “perspectives”.

(2-6) Please make conclusions focused on the subject and “perspectives”.

(2-7) Please consider adding one paragraph describing the future perspectives.

(2-8) Please remove information in section Supplementary Materials since documents do not exist.

Good luck and all the best J

Author Response

C/o Editorial Office

Pharmaceuticals MDPI

April 12, 2023

Dear Editorial Office:

The submitted manuscript entitled Cardiovascular safety and benefits of testosterone implant therapy in postmenopausal women: where are we? by Renke et al. has been re-evaluated and has undergone revisions requested and suggested by the reviewers. The manuscript and new tables were sent with the corrections and adjustments flagged as "revision" in the Word reviewer.

The authors would like to respond to all reviewers' comments as follows promptly:

REVIEWERS 2 COMMENTS

(1) ABSTRACT

Please rewrite the text of the abstract to be more focus on the subject/the title. At the present form, only last sentence describes the idea of the manuscript. All other sentences describe very well-known findings. Please consider to start the abstract with the aim of the study.

A1: We thank the reviewer for drawing attention to this point and for the opportunity to clarify it. We agreed with the reviewer and adjusted the abstract: We rewrote the abstract according to the reviewer's request.

(2) GENERAL

(2-1) The significant amount of the text describes the very well-known and general knowledge. Please focus on the subject. Please consider rewriting the text to focus on “perspectives”; not to very well-known knowledge. It will increase interest of readers and “visibility” of the manuscript.

A2-1: We thank the reviewer for drawing attention to this point and for the opportunity to clarify it. We agree with the reviewer, and we rewrote a large part of the manuscript with an emphasis on new concepts and perspectives of testosterone implant therapy as recommended by the reviewer.

(2-2) Please cite the original (instead of review), and important pioneered results, as well as recent advance in the field focusing on the subject of the study. The manuscript is aimed to be “perspective”. At the present form it is not.

A2-2: We thank the reviewer for drawing attention to this point and for the opportunity to clarify it. We agree with the reviewer, and we added the subsection “1.8.New Directions and future approaches in STT” with pioneered results and also proposed the criteria (IDEALSTT) that we consider innovative according to our great clinical expertise and previous trials (table 2).

(2-3) Please list rules of all related agencies (not only FDA and Brazilian).

A2-3: We thank the reviewer for drawing attention to this point and for the opportunity to clarify it. We agree with the reviewer and added the entities' rules/positions (Endocrine Society in the US and SBEM (Brazilian Society of Endocrinology and Metabolism) in Brazil).

(2-4) Please be precise in the description. One of the examples is “estrogen” (line 179: “The cardioprotective effect from estrogen and possible deleterious action from T on the…”). It is very well known that there are three main estrogens (estradiol, estriol, estrone). Accordingly, please replace “estrogen” with “estrogens”. Please go through the text and please be precise in the description.

A2-4: We thank the reviewer for drawing attention to this point and for the opportunity to clarify it. We agree with the reviewer, and we adjusted the term “estrogens” with the accurate description of “estrogen”or “estradiol”.

(2-5) Please remove Figure 1 since it presents very well knowledge. Please remove the text related to Figure 1. Please keep in mind that you chose to write “perspectives”.

A2-5: We thank the reviewer for drawing attention to this point and for the opportunity to clarify it. We disagree with the reviewer. Figure 1 demonstrates the procedure of silastic implant insert that is not well-known by many physicians around the world, and we consider it essential for the perspective.

(2-6) Please make conclusions focused on the subject and “perspectives”.

A2-6: We thank the reviewer for drawing attention to this point and for the opportunity to clarify it. We agree with the reviewer and adjusted the conclusion focused on the perspective.

(2-7) Please consider adding one paragraph describing the future perspectives.

A2-7: We thank the reviewer for drawing attention to this point and for the opportunity to clarify it. We agree with the reviewer and adjusted the text as described in answer A2-2.  

(2-8) Please remove information in section Supplementary Materials since documents do not exist.

A2-8: We thank the reviewer for drawing attention to this point and for the opportunity to clarify it. We agree with the reviewer and we excluded this section.

We hope you find our manuscript suitable for publication and look forward to hearing from you in due course. We are gratefully and appreciate the reviewers' suggestions and remain available for further adjustments and corrections.

Sincerely,

Guilherme Renke MD

Tel: +5521994431653

Email: renke@renke.com.br

Round 2

Reviewer 1 Report

No

Author Response

We thank the reviewer for drawing attention to this point and for the opportunity to clarify it. We agreed with the reviewer and adjusted the manuscript and performed a fine/minor spell check. 

Best Regards,

GR

Reviewer 2 Report

Pharmaceuticals

COMMENTS TO THE EDITOR AND THE AUTHORS

Manuscript ID 2289239R1: “Cardiovascular safety and benefits of testosterone implant therapyin postmenopausal women: where are we?"

Dear Editor and Authors,

The authors improved manuscript to some extent, but the manuscript has not been sufficiently improved to warrant publication in Pharmaceuticals. Some important issues are not improved.

Please find below some of the comments.

COMMENTS

(1) ABSTRACT

Although authors rewrite the abstract, they did not rewrite it correctly. Please rewrite the text of the abstract to be more focus on the subject/the title. At the present form, only last sentence describes the idea of the manuscript. All other sentences describe very well-known findings. Please consider starting the abstract with the aim of the study  (line 17: Here, we discuss the …..) and then describe the content of the review.

(2) GENERAL

(2-1) Please remove Figure 1 since it presents very well knowledge. Please remove the text related to Figure 1. Please keep in mind that you chose to write “perspectives”. At XXI century and very well applied AI Figure 1 is not “perspectives”.

(2-2) Please remove files named “supplementary” and “not published materials” since they are included in the main file of the manuscript.

Good luck and all the best ?

Author Response

(1) ABSTRACT

Although authors rewrite the abstract, they did not rewrite it correctly. Please rewrite the text of the abstract to be more focus on the subject/the title. At the present form, only last sentence describes the idea of the manuscript. All other sentences describe very well-known findings. Please consider starting the abstract with the aim of the study  (line 17: Here, we discuss the …..) and then describe the content of the review.

"We thank the reviewer for drawing attention to this point and for the opportunity to clarify it. We agreed with the reviewer and we rewrote the abstract according to the reviewer's request."

(2) GENERAL

(2-1) Please remove Figure 1 since it presents very well knowledge. Please remove the text related to Figure 1. Please keep in mind that you chose to write “perspectives”. At XXI century and very well applied AI Figure 1 is not “perspectives”.

"We thank the reviewer for drawing attention to this point and for the opportunity to clarify it. We agreed with the reviewer and we removed Figure 1 and the text related to it."

(2-2) Please remove files named “supplementary” and “not published materials” since they are included in the main file of the manuscript.

"We thank the reviewer for drawing attention to this point and for the opportunity to clarify it. We agreed with the reviewer and we removed all supplementary file". 

Thank you and Best Regards!!

GR
